# Inferring a simple mechanism for alpha-blocking by fitting a neural population model to EEG spectra

**Agus Hartoyo**[1]*, **Peter J. Cadusch**[2], **David T. J. Liley**[3,4], **Damien G. Hicks**[1,2,5]*

**1** Optical Sciences Centre, Swinburne University of Technology, Hawthorn, Victoria, Australia, **2** Department of Physics and Astronomy, Swinburne University of Technology, Hawthorn, Victoria, Australia, **3** Centre for Human Psychopharmacology, School of Health Sciences, Swinburne University of Technology, Hawthorn, Victoria, Australia, **4** Department of Medicine, University of Melbourne, Parkville, Victoria, Australia, **5** Bioinformatics Division, Walter & Eliza Hall Institute of Medical Research, Parkville, Victoria, Australia

* ahartoyo@swin.edu.au (AH); dghicks@swin.edu.au (DGH)

**Data Availability Statement:** All EEG data files are available from the PhysioNet database (https://archive.physionet.org/pn4/eegmmidb/). The implementation of the methods and all datasets are

## Abstract

Alpha blocking, a phenomenon where the alpha rhythm is reduced by attention to a visual, auditory, tactile or cognitive stimulus, is one of the most prominent features of human electroencephalography (EEG) signals. Here we identify a simple physiological mechanism by which opening of the eyes causes attenuation of the alpha rhythm. We fit a neural population model to EEG spectra from 82 subjects, each showing a different degree of alpha blocking upon opening of their eyes. Though it has been notoriously difficult to estimate parameters by fitting such models, we show how, by regularizing the differences in parameter estimates between eyes-closed and eyes-open states, we can reduce the uncertainties in these differences without significantly compromising fit quality. From this emerges a parsimonious explanation for the spectral differences between states: Changes to just a single parameter, $p_{ei}$, corresponding to the strength of a tonic excitatory input to the inhibitory cortical population, are sufficient to explain the reduction in alpha rhythm upon opening of the eyes. We detect this by comparing the shift in each model parameter between eyes-closed and eyes-open states. Whereas changes in most parameters are weak or negligible and do not scale with the degree of alpha attenuation across subjects, the change in $p_{ei}$ increases monotonically with the degree of alpha blocking observed. These results indicate that opening of the eyes reduces alpha activity by increasing external input to the inhibitory cortical population.

## Author summary

One of the most striking features of the human electroencephalogram (EEG) is the presence of neural oscillations in the range of 8-13 Hz. It is well known that attenuation of these alpha oscillations, a process known as alpha blocking, arises from opening of the eyes, though the cause has remained obscure. In this study we infer the mechanism underlying alpha blocking by fitting a neural population model to EEG spectra from 82 different

publicly available at https://github.com/cds-swinburne/Hartoyo-et-al-2020-DATA-n-CODE.

**Funding:** This work was supported in part by a Swinburne Postgraduate Research Award to AH and in part by an Australian Research Council (https://www.arc.gov.au/) grant FT140101104 to DGH. Computations were performed on the gSTAR/ozSTAR national facilities at Swinburne University of Technology funded by Swinburne and the Australian Government's Education Investment Fund. The funders had no role in study design, data collection and analysis, decision to publish, or preparation of the manuscript.

**Competing interests:** The authors have declared that no competing interests exist.

individuals. Although such models have long held the promise of being able to relate macroscopic recordings of brain activity to microscopic neural parameters, their utility has been limited by the difficulty of inferring these parameters from fits to data. Our approach involves fitting eyes-open and eyes-closed EEG spectra in a way that minimizes unnecessary differences in model parameters between the two states. Surprisingly, we find that changes in just one parameter, the level of external input to the inhibitory neurons in cortex, is sufficient to explain the attenuation of alpha oscillations. This indicates that opening of the eyes reduces alpha activity simply by increasing external inputs to the inhibitory neurons in the cortex.

## Introduction

Alpha blocking is a classic feature of the human electroencephalogram (EEG). First identified by Hans Berger as part of his discovery of human EEG in the 1920's [1, 2], it is now arguably its most robust empirical feature. Classically, alpha blocking refers to the reduction in spontaneously-recorded occipital alpha band (8-13 Hz) power in response to opening of the eyes [3]. More generally, changes in alpha-band power can be effected by a range of visual, tactile and auditory stimuli and altered states of arousal and is widely used as a diagnostic of cognitive activity [4–7].

Despite the importance of alpha blocking in studies of cognition, it still lacks a generally-accepted, mechanistic understanding [8]. Importantly, the mechanism associated with alpha blocking is typically considered separately from the mechanism associated with alpha wave generation. Whereas cortical alpha is often thought to be generated by feedforward and feedback interactions between the thalamus and overlying cortex [9–12], blocking is considered to arise from changes in the phase synchrony of populations of these near-identical cortico-thalamic alpha oscillators [13–15]. In this paper, we show how alpha generation and blocking can be described self-consistently within a single neural population model for the cortex.

Neural population models describe how microscopic properties in the cortex, such as postsynaptic rate constants, affect macroscopic observables, such as the local field potential detected by the EEG [16–18]. These models match the high time resolution and low spatial resolution of the EEG and have long been used to interpret the characteristics of alpha-band activity [12, 19, 20]. Notably, it has been shown how, with judiciously chosen model parameter values, alpha oscillations can arise spontaneously in the cortex [13, 21–26], without the need for direct pacing by oscillatory inputs [27–30]. However, it has been more difficult to interpret alpha blocking within these models since there are multiple ways to reduce or eliminate alpha activity [31]. For example, alpha attenuation has been attributed to coincident changes in several thalamo-cortical parameters controlling the feedforward, cortico-thalamo-cortical, and intra-cortical circuits [32].

A fundamental challenge in using neural population models is the difficulty in estimating parameter values directly from fits to EEG data [33]. Although forward calculations have provided plausible explanations for spontaneous alpha generation, solving the inverse problem to determine the many unknown model parameters is crucial if we want to relate the subject-to-subject variability observed in EEG signals to an associated variability in specific microscopic parameters. Achieving this will help identify (and potentially control) the underlying microscopic drivers of the EEG response, and associated cognitive behavior, in a given individual.

Recently [33], we examined the large parameter uncertainties associated with fitting a neural population model [22, 26] to EEG data. These large (and correlated) uncertainties mean

that model parameters remain mostly unconstrained even though the data are fit accurately. This problem is referred to as model unidentifiability [34, 35] and sloppiness [36, 37] and is typical when fitting models with many parameters. Nevertheless, our study found that one out of the 22 parameters was individually identifiable. This single identifiable parameter, the decay rate of the inhibitory post-synaptic potential $\gamma_i$, was discovered by fitting to EEG spectra from subjects with their eyes closed. The value of this inhibitory decay rate needed to be within a narrow range in order to generate alpha oscillations (of any amplitude) from a white noise input, regardless of the values of the other parameters. This demonstrated the fundamental importance of this parameter in generating spontaneous alpha-band activity.

To understand alpha blocking, however, we must confront another aspect of the unidentifiability problem: whether one can learn the *change* in a parameter in response to a particular stimulus, in this case the opening of the eyes. This is important since it is often more useful to know how much a parameter changes in response to a stimulus than it is to know the absolute value of that parameter. We refer to this as the 2-state fitting problem since this will involve fitting two spectra (eyes closed and eyes open) from a single individual. Thus our previous study [33], where we only fit to the eyes-closed spectra in each individual, was a 1-state fitting problem.

Naively, it would seem that the unidentifiability we found for the 1-state problem would doom the 2-state fitting problem since one seemingly needs to perform separate fits to each state. However, by fitting the two states simultaneously and by penalizing parameter differences between the states, we are able to reliably determine the change, or differential response, of a particular parameter, even though the absolute value of that parameter in each state can be quite uncertain. When examining data across many subjects, we are able to associate a single parameter $p_{ei}$—the strength of extra-cortical input to the inhibitory cortical population—with the attenuation of alpha oscillations upon opening of the eyes. This unifies the mechanisms for alpha generation and blocking within a single model.

In the rest of this introduction we briefly describe the data, the model, and the fitting strategy. Further details about methodology are given in the "Methods" section.

## EEG data

The EEG data used in this study is provided in the online repository [38] (https://archive.physionet.org/pn4/eegmmidb/). We use data from the occipital electrode from 82 individuals, as in our previous study [33], although this time we use eyes-open as well as eyes-closed data. We apply Welch's method [39] to estimate the $2 \times 82$ power spectra. Once again, because of the well-known nonlinearities and nonstationarities in EEG recordings, we restrict our study to frequencies between 2 Hz and 20 Hz. Since the absolute power in the EEG data is not meaningful, each spectrum is normalized to have a total power of 1. Our interest is thus in changes in spectral shape, not magnitude, upon going from eyes closed to eyes open states in each individual.

**EEG data variability across individuals.**   It is well-known that the degree to which the alpha rhythm is attenuated by a given visual stimulus varies across individuals [40] exhibiting, for example, a negative correlation with age [41]. Inspection of the spectra we use from the 82 subjects shows that there is substantial variability in the degree of alpha blocking across individuals used in this study (see Fig 1 for a sample set of spectra, Fig A S1 Appendix for the full set).

Our approach is to use this individual variability to quantify how much each parameter shifts between EC to EO states and how these shifts scale with the degree of alpha blocking. To do this quantitatively, we needed to define a measure of alpha blocking strength. Here we use

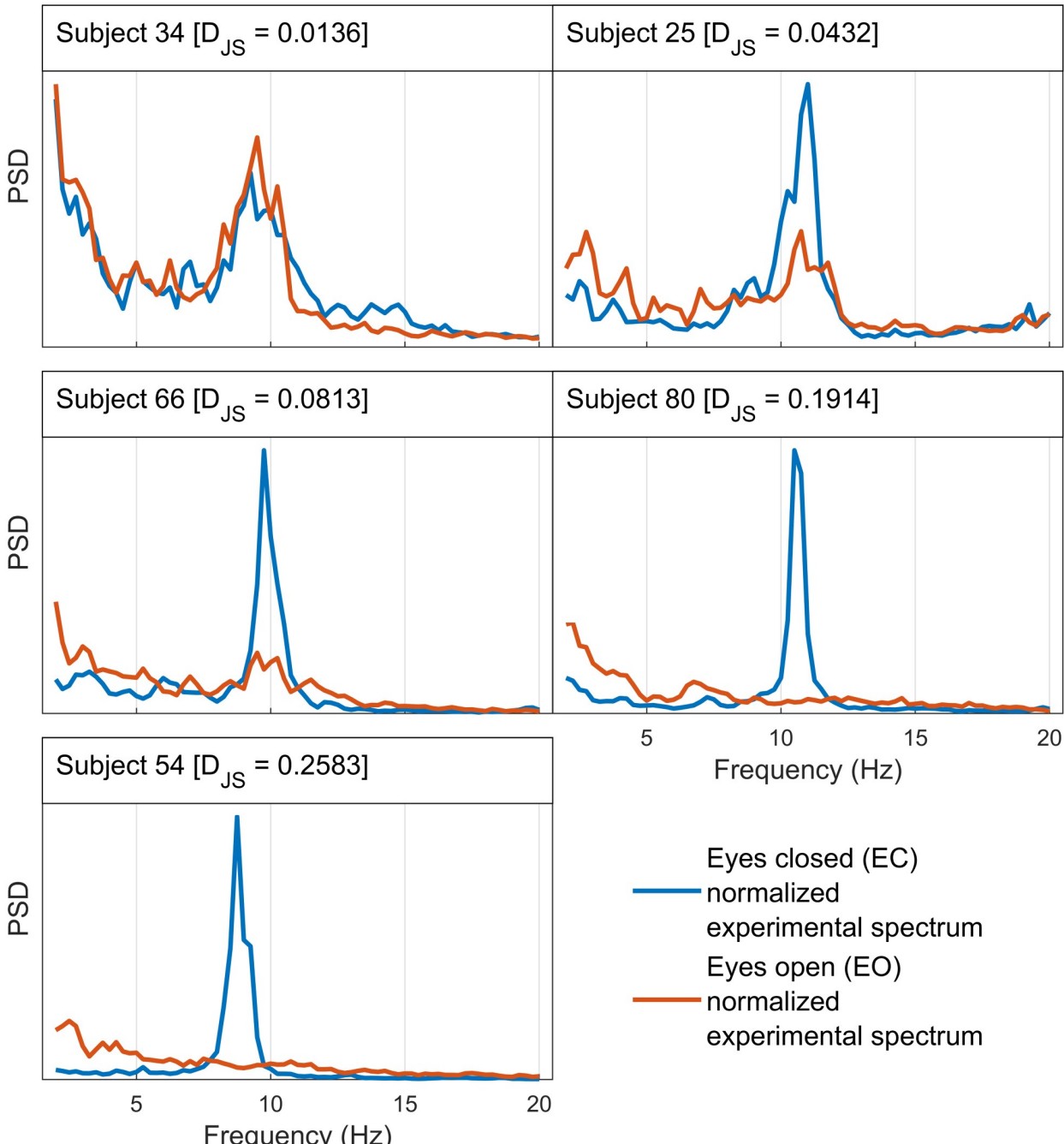

**Fig 1. Different subjects exhibit different degrees of alpha blocking upon opening of the eyes.** Here five subjects have been selected to illustrate the range of alpha blocking behaviour observed in the dataset. The vertical axis on each plot represents an arbitrary scale for the normalized power spectral density (PSD). Some subjects do not show any reduction in alpha power between EC and EO states (e.g. Subject 34); others exhibit partial blocking where the alpha activity in EO state is weaker than that of EC but is still pronounced (e.g. Subject 25); while some show total blocking where the alpha activity in the EO spectra completely disappears (e.g. Subject 80). To quantify the degree to which the EEG spectrum changes upon opening of the eyes, we compute the Jensen-Shannon divergence, $D_{JS}$, between the eyes-closed (EC) and eyes-open (EO) normalized experimental spectrum for each subject. A larger value of $D_{JS}$ implies more pronounced EEG spectrum changes, or alpha-wave suppression. The complete set of spectra for all subjects is presented in Fig A in S1 Appendix, ordered by $D_{JS}$.

the Jensen-Shannon divergence, $D_{JS}$, which provides a scalar measure of the difference in shape between (normalized) eyes-closed (EC) and eyes-open (EO) spectra from each individual (see Section "Jensen-Shannon divergence as a measure of the degree of alpha blocking"). To demonstrate how this measure aligns with our intuitive notion of spectrum change, the EC and EO data in Fig 1 are ordered by increasing $D_{JS}$. Although we explored alternative measures such as the change in relative strength of the alpha band component, we use $D_{JS}$ since it is a more global measure of function change that does not rely on defining a particular frequency band. As we will show, by comparing how parameter differences scale with increasing $D_{JS}$ we are able to establish how much each (microscopic) parameter changes with the degree of alpha-blocking.

## Neural population model

The model used in this paper is the local variant of the mean-field model originally described in Refs [22, 26]. As described in our previous study [33], this model consists of a coupled set of first and second order non-linear ordinary differential equations parameterized by 22 physiologically-motivated parameters (see Table 1) as presented below:

$$\tau_e \frac{dh_e(t)}{dt} = h_e^{rest} - h_e(t) + \frac{h_e^{eq} - h_e}{|h_e^{eq} - h_e^{rest}|} I_{ee}(t) + \frac{h_i^{eq} - h_e}{|h_i^{eq} - h_e^{rest}|} I_{ie}(t), \tag{1}$$

$$\tau_i \frac{dh_i(t)}{dt} = h_i^{rest} - h_i(t) + \frac{h_e^{eq} - h_i}{|h_e^{eq} - h_i^{rest}|} I_{ei}(t) + \frac{h_i^{eq} - h_i}{|h_i^{eq} - h_i^{rest}|} I_{ii}(t), \tag{2}$$

$$\frac{d^2 I_{ee}(t)}{dt^2} + 2\gamma_e \frac{dI_{ee}(t)}{dt} + \gamma_e^2 I_{ee}(t) = \Gamma_e \gamma_e e(N_{ee}^\beta S_e(h_e) + p_{ee}(t)), \tag{3}$$

$$\frac{d^2 I_{ie}(t)}{dt^2} + 2\gamma_i \frac{dI_{ie}(t)}{dt} + \gamma_i^2 I_{ie}(t) = \Gamma_i \gamma_i e N_{ie}^\beta S_i(h_i), \tag{4}$$

$$\frac{d^2 I_{ei}(t)}{dt^2} + 2\gamma_e \frac{dI_{ei}(t)}{dt} + \gamma_e^2 I_{ei}(t) = \Gamma_e \gamma_e e(N_{ei}^\beta S_e(h_e) + p_{ei}(t)), \tag{5}$$

$$\frac{d^2 I_{ii}(t)}{dt^2} + 2\gamma_i \frac{dI_{ii}(t)}{dt} + \gamma_i^2 I_{ii}(t) = \Gamma_i \gamma_i e N_{ii}^\beta S_i(h_i), \tag{6}$$

where

$$S_j(h_j) = \frac{S_j^{max}}{\left(1 + \exp\left(-\sqrt{2}\frac{(h_j - \bar{\mu}_j)}{\sigma_j}\right)\right)}; \quad j = e, i. \tag{7}$$

Local equations are linearized around a fixed point and the power spectral density (PSD) is derived assuming a stochastic driving signal of the excitatory population that represents thalamo-cortical and long range cortico-cortical inputs, assumed to be Gaussian white noise. The modelled PSD can then be written as a rational function of frequency derived from the transfer function for the linearized system. As was explained in earlier studies [33, 42], tonic excitatory signals to the inhibitory ($p_{ei}$) and excitatory($p_{ee}$) populations are included as unknown parameters to account for potential constant offsets in extracortical inputs.

**Table 1. State-distinct parameters and state-common parameters.**

| Type | Physiological parameters | | | | Fitting parameters | |
|---|---|---|---|---|---|---|
| | No | Label | Description | Interval | No | Label |
| State-distinct parameters | 1 | $\tau_e$ | Passive membrane decay time const. of the excitatory population | [5, 150] ms | 1 | $\tau_e(EC)$ |
| | | | | | 2 | $\tau_e(EO)$ |
| | 2 | $\tau_i$ | Passive membrane decay time const. of the inhibitory population | [5, 150] ms | 3 | $\tau_i(EC)$ |
| | | | | | 4 | $\tau_i(EO)$ |
| | 3 | $\gamma_e$ | Excitatory postsynaptic potential rate constant | [0.1, 1.0] /ms | 5 | $\gamma_e(EC)$ |
| | | | | | 6 | $\gamma_e(EO)$ |
| | 4 | $\gamma_i$ | Inhibitory postsynaptic potential rate constant | [0.01, 0.1] /ms | 7 | $\gamma_i(EC)$ |
| | | | | | 8 | $\gamma_i(EO)$ |
| | 5 | $\Gamma_e$ | Postsynaptic potential amplitude of the excitatory population | [0.1, 2.0] mV | 9 | $\Gamma_e(EC)$ |
| | | | | | 10 | $\Gamma_e(EO)$ |
| | 6 | $\Gamma_i$ | Postsynaptic potential amplitude of the inhibitory population | [0.1, 2.0] mV | 11 | $\Gamma_i(EC)$ |
| | | | | | 12 | $\Gamma_i(EO)$ |
| | 7 | $p_{ee}$ | Rate of the excitatory input to the excitatory population | [0.0, 10.0] /ms | 13 | $p_{ee}(EC)$ |
| | | | | | 14 | $p_{ee}(EO)$ |
| | 8 | $p_{ei}$ | Rate of the excitatory input to the inhibitory population | [0.0, 10.0] /ms | 15 | $p_{ei}(EC)$ |
| | | | | | 16 | $p_{ei}(EO)$ |
| | 9 | $\eta$ | Exponent of the input spectrum | [0.0, 2.0] | 17 | $\eta(EC)$ |
| | | | | | 18 | $\eta(EO)$ |
| State-common parameters | 10 | $h_e^{rest}$ | Mean resting membrane potential of the excitatory population | [-80, -60] mV | 19 | $h_e^{rest}(EC, EO)$ |
| | 11 | $h_i^{rest}$ | Mean resting membrane potential of the inhibitory population | [-80, -60] mV | 20 | $h_i^{rest}(EC, EO)$ |
| | 12 | $h_e^{eq}$ | Mean Nernst membrane potential of the excitatory population | [-20, 10] mV | 21 | $h_e^{eq}(EC, EO)$ |
| | 13 | $h_i^{eq}$ | Mean Nernst membrane potential of the inhibitory population | [-90, -65] mV ‡ | 22 | $h_i^{eq}(EC, EO)$ |
| | 14 | $S_e^{max}$ | Maximum mean firing rate of the excitatory population | [0.05, 0.5] /ms | 23 | $S_e^{max}(EC, EO)$ |
| | 15 | $S_i^{max}$ | Maximum mean firing rate of the inhibitory population | [0.05, 0.5] /ms | 24 | $S_i^{max}(EC, EO)$ |
| | 16 | $\bar{\mu}_e$ | Firing thresholds of the excitatory population | [-55, -40] mV | 25 | $\bar{\mu}_e(EC, EO)$ |
| | 17 | $\bar{\mu}_i$ | Firing thresholds of the inhibitory population | [-55, -40] mV | 26 | $\bar{\mu}_i(EC, EO)$ |
| | 18 | $\sigma_e$ | Std. deviation of firing thresholds of the excitatory population | [2, 7] mV | 27 | $\sigma_e(EC, EO)$ |
| | 19 | $\sigma_i$ | Std. deviation of firing thresholds of the inhibitory population | [2, 7] mV | 28 | $\sigma_i(EC, EO)$ |
| | 20 | $N_{ee}^{\beta}$ | # of connections an excitatory neuron receives from excitatory neurons | [2000, 5000] | 29 | $N_{ee}^{\beta}(EC, EO)$ |
| | 21 | $N_{ei}^{\beta}$ | # of connections an inhibitory neuron receives from excitatory neurons | [2000, 5000] | 30 | $N_{ei}^{\beta}(EC, EO)$ |
| | 22 | $N_{ie}^{\beta}$ | # of connections an excitatory neuron receives from inhibitory neurons | [100, 1000] | 31 | $N_{ie}^{\beta}(EC, EO)$ |
| | 23 | $N_{ii}^{\beta}$ | # of connections an inhibitory neuron receives from inhibitory neurons | [100, 1000] | 32 | $N_{ii}^{\beta}(EC, EO)$ |

The model is characterized by 23 physiological parameters associated with a given subject. As the subject moves from the EC state to the EO state, so do the physiological parameters. A *state-distinct parameter* is a physiological parameter that changes between states and corresponds to two distinct fitting-parameters. A *state-common parameter* is kept the same for both the EC and EO states and corresponds to a single fitting-parameter. There are 9 state-distinct parameters (which translate into twice as many fitting parameters) and 14 state-common parameters giving a total of 32 adjustable parameters to optimize during the joint fitting to both spectra for each individual. Minimum and maximum values for the physiological parameters are presented. The list of the physiologically-plausible intervals was originally proposed in [42], and is here updated with a reduced interval for $\gamma_i$ as suggested by the identifiability analysis conducted in [33].

‡The physiologically-plausible interval for $h_i^{eq}$ presented in this table corrects a typographical error made in [33] which incorrectly indicated the parameter's minimum and maximum to be -20 mV and 10 mV, respectively.

In this study, we use the identical model but with two changes. The first is to introduce an additional parameter to allow for a non-white background spectrum (giving a total of 23 parameters—see Table 1). Though this adds an extra degree of freedom it is necessary in order to achieve fits to some of the eyes-open spectra. In fact there is evidence from EEG and ECoG

studies (for example, see [43]) that the background PSD may have a frequency dependence (typically quoted as $1/f$) not readily accounted for by a rational transfer function alone. While various approaches have been suggested to account for such a dependence, we have chosen the simplest way to incorporate it into our model by relaxing the white noise assumption and using coloured noise for the driving signal. Specifically, we take the input PSD, $S^{in} \propto 1/f^{\eta}$ where $\eta$ is the exponent of the input spectrum treated as a new state-dependent adjustable parameter in the range $0 \leq \eta \leq 2$; $\eta = 0$ corresponds to the original white noise, $\eta = 1$ to the pure $1/f$ (pink) noise, and $\eta = 2$ to a Wiener process (Brownian noise).

The second change is to incorporate the main result learned from [33] and restrict the range of $\gamma_i$. There it was found that the inhibitory rate constant $\gamma_i$ has a sharply peaked posterior distribution, making it (uniquely) identifiable in the eyes-closed case. This was reproduced here in the eyes-open data when the EEG had a detectable peak in the alpha band; if no peak was observed, the posterior distribution resembled the assumed prior distribution. In light of this and in line with the search for a parsimonious explanation for alpha blocking, the prior distributions for the eyes-closed and eyes open cases in the current study were both limited to a reduced interval around the range found for its posterior distribution in [33] (See the updated minimum and maximum value for $\gamma_i$ in Table 1).

## Model fitting strategy

In this 2-state fitting problem, the EC spectrum and the EO counterpart from a given subject are treated as a single dataset to be jointly fit by the model. Given that a single spectrum fit has 23 unknown parameters, a naive fit to two spectra would have 46 potentially unknown parameters.

To reduce the number of unknowns we implement two types of constraint. The first constraint (see Section "State-common parameters and state-distinct parameters") is that 14 of the parameters should remain the same in both the EC and EO conditions. This set is referred to as *state-common parameters*. The remaining 9 parameters are allowed to vary between conditions and are thus referred to as *state-distinct parameters*, giving a total of 32 unknown parameters. The list of parameters belonging to both types is presented in Table 1. This joint-fitting approach for the two spectra allows us to couple together the dependency between the EC and EO parameters while at the same time allowing their actual values to be determined by the data.

The second constraint (see Section "Regularization of parameter differences") is to penalize (regularize) non-zero differences between EC and EO values for state-distinct parameters. This helps to identify the important parameter differences driving the change in spectral shape from EC to EO. Our regularization procedure is a variant on the standard procedure employed in high-dimensional inference problems searching for sparse, or parsimonious, solutions [44].

We use the same fitting scheme to that described in [33]: Fitted parameters are obtained using particle swarm optimization (PSO) [45, 46] starting from a random set of initial states. Each of the 82 subjects was fit separately as a parallel job on the OzStar supercomputer at Swinburne University of Technology, generating 1000 independent fit samples per subject. Computations were performed using a parallel for-loop with 30 workers and 30 CPUs each with 1 GB of memory. From the resulting sample of 1000 optimized parameter sets, the 10 percent with the lowest cost function values are accepted as final estimates (a detailed discussion justifying this threshold was given previously [33]).

Further details on data analysis are given in Section "Methods". Our implementation of the methods and all datasets are publicly available at https://github.com/cds-swinburne/Hartoyo-et-al-2020-DATA-n-CODE.

## Results

Fig 2 shows the best model fits to EC and EO spectra from 5 different subjects, ordered verti-cally by degree of alpha blocking. Both regularized and unregularized cases exhibit good fits to the data. The similarity between regularized and unregularized cases confirms that the bias caused by regularization is within acceptable limits.

The EC and EO posterior marginal distributions for each parameter are shown in Fig 3. Plots for the 5 subjects are ordered vertically by degree of alpha blocking, as in Fig 2. Distribu-tions are estimated from the 100 best fit parameter sets for each subject. $p_{ei}$ shows the most noticeable difference between its EC and EO distributions, with EO distributions drifting increasingly higher than their corresponding EC distributions as alpha blocking gets larger. Differences between distributions for EC and EO states are weakly visible for $p_{ee}$ and mostly negligible for other parameters.

To better quantify the difference between EC and EO states for each parameter and how it scales with the degree of alpha blocking, we calculate the difference between each EC to EO parameter estimate. We do this for each of the $N_J = 100$ best sample fits found for each of the $N_I = 82$ subjects. Thus, if $\theta_{ij}$ is a given parameter estimate indexed by subject, $i$, and sample fit, $j$, we define the *parameter response*, $\Delta\theta_{ij}$, to be

$$\Delta\theta_{ij} = \theta_{ij}^{\text{EO}} - \theta_{ij}^{\text{EC}}, \qquad i \in \{1, \ldots, N_I\}, \quad j \in \{1, \ldots, N_J\} \tag{8}$$

$$\overline{\Delta\theta_i} = \frac{1}{N_J} \sum_{j=1}^{N_J} \left( \theta_{ij}^{\text{EO}} - \theta_{ij}^{\text{EC}} \right) \tag{9}$$

where $\overline{\Delta\theta_i}$ is the resulting mean parameter response from EC to EO (averaged over sample fits) for a given subject, $i$.

In Fig 4, to examine the association between each parameter response and the degree of alpha blocking, we plot $\overline{\Delta\theta_i}$ versus $D_{JS}^i$ for each of the 82 subjects $i$. We also plot the 25% to 75% interquartile range determined from that subject's $N_J = 100$ sample fits, which provides an estimate of the unidentifiability of the parameter response. Results are shown for all 9 state-dis-tinct parameters.

To characterize how each parameter response scales with the degree of alpha blocking, we perform a linear regression of $\Delta\theta$ versus $D_{JS}$. We use linearity simply to characterize the trend, not because of any expectation of linearity. Most parameter responses are either zero or show an insignificant trend with the degree of alpha blocking. The major exception is $\Delta p_{ei}$ which increases monotonically with increasing $D_{JS}$. $p_{ee}$ also shows a non-zero parameter response, although its trend with $D_{JS}$ is weak and restricted to low values. In the context of our model this implies that excitatory input to the inhibitory population is the dominant factor determin-ing the response of alpha oscillations to a visual stimulus.

## Discussion

By fitting a neural population model to EEG data from 82 individuals, we have demonstrated a clear association between the degree of alpha blocking and a single model parameter, $p_{ei}$: the strength of a tonic excitatory input to the inhibitory population. Most of the change between eyes-closed and eyes-open spectra is explained by variation in this external input level. This single-parameter explanation for the difference between eyes-closed and eyes-open spectra contrasts with previous explanations for alpha blocking which invoked changes in multiple parameters [31, 32].

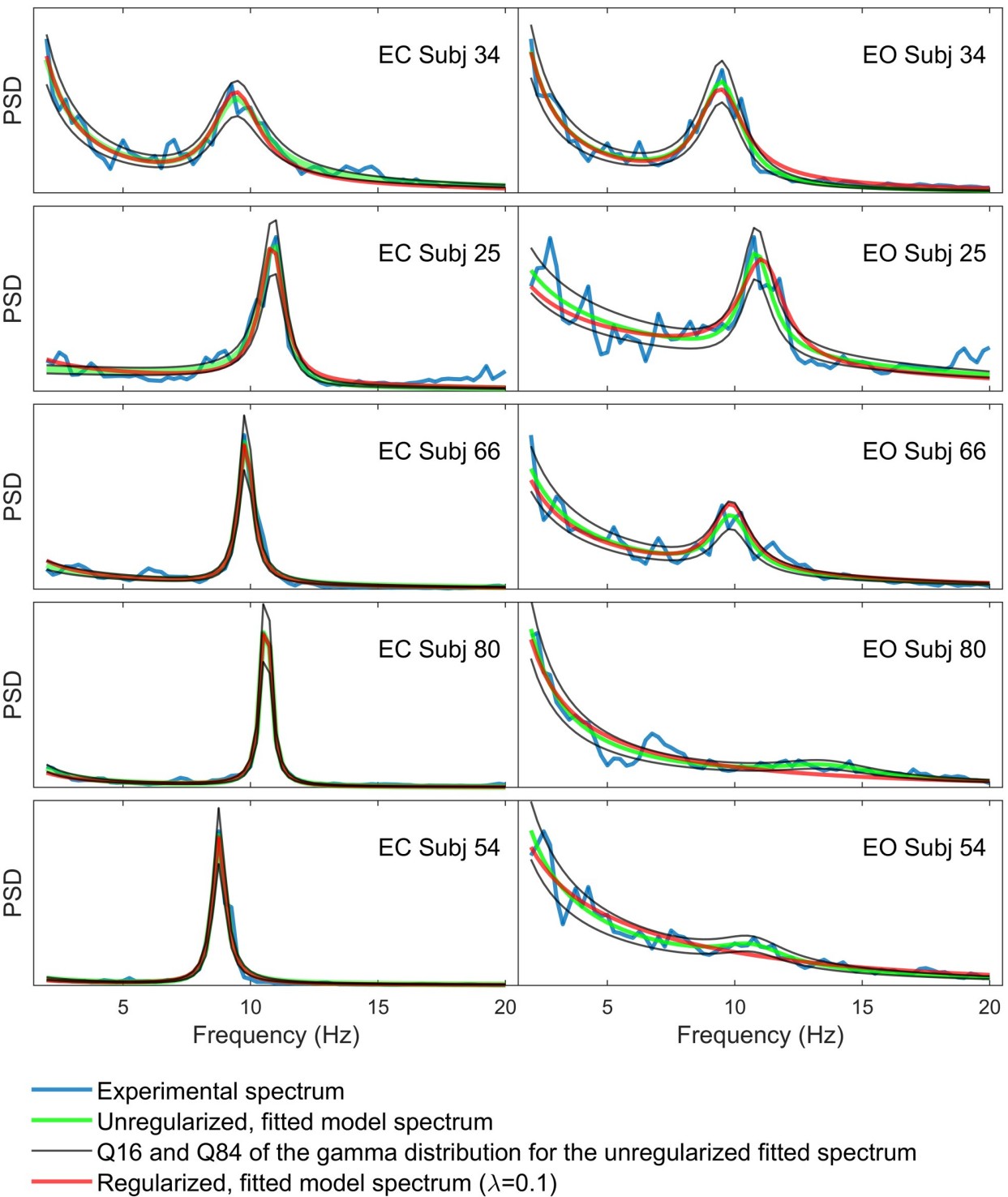

**Fig 2. Regularized and unregularized best fits to EC and EO spectra.** Best fit results for the 5 subjects shown in Fig 1. Subjects are ordered vertically by the degree of alpha blocking, with alpha blocking increasing downwards. Regularized fits (red) deviate only slightly from the unregularized fits (green). The 16% and 84% uncertainty quantiles (based on the gamma distribution for the unregularized best fits) are shown in black. These boundaries define the acceptable error of a fit. Regularized best fits deviate only slightly from the unregularized ones and generally stay within these uncertainty quantiles. In order to visualize the different fits, EC and EO spectra for a given subject are not necessarily shown on the same vertical scale.

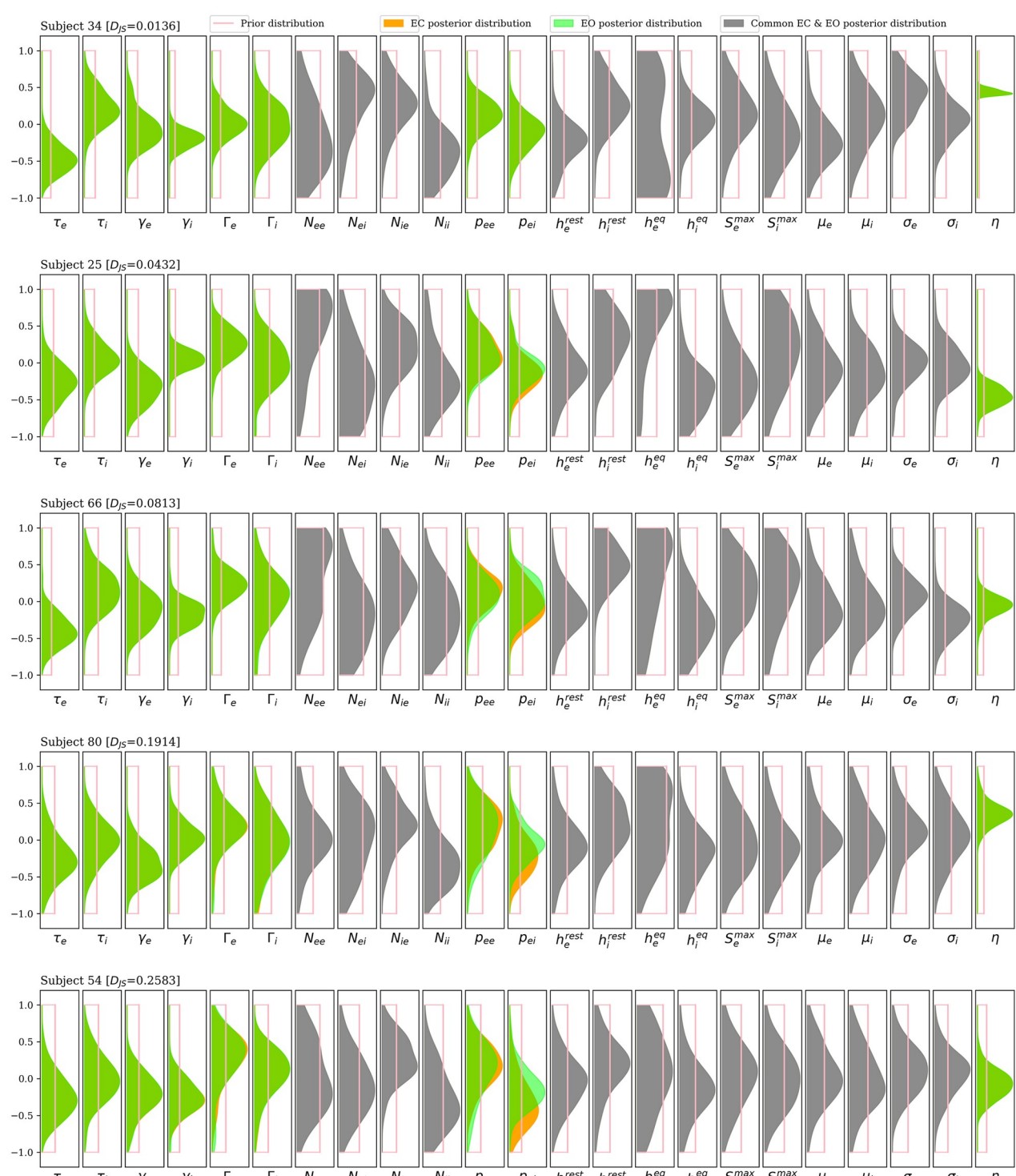

**Fig 3. Posterior distributions for each parameter.** Posterior distributions for state-distinct parameters (with EC in orange and EO in green) and state-common parameters (grey), again for the 5 subjects in Figs 1 and 2. Subjects are ordered vertically by the degree of alpha blocking, with alpha blocking increasing downwards. The distributions are calculated using kernel density estimates from the best 100 of 1000 randomly seeded particle swarm optimizations for each subject. Each parameter is plotted in normalized coordinates, where -1 corresponds to the lower limit of the plausible parameter interval and +1 corresponds to the upper limit. The parameter $p_{ei}$ is the only parameter where the difference between EC and EO distributions increases consistently with the degree of alpha blocking. Weaker shifts in $p_{ee}$ are also apparent.

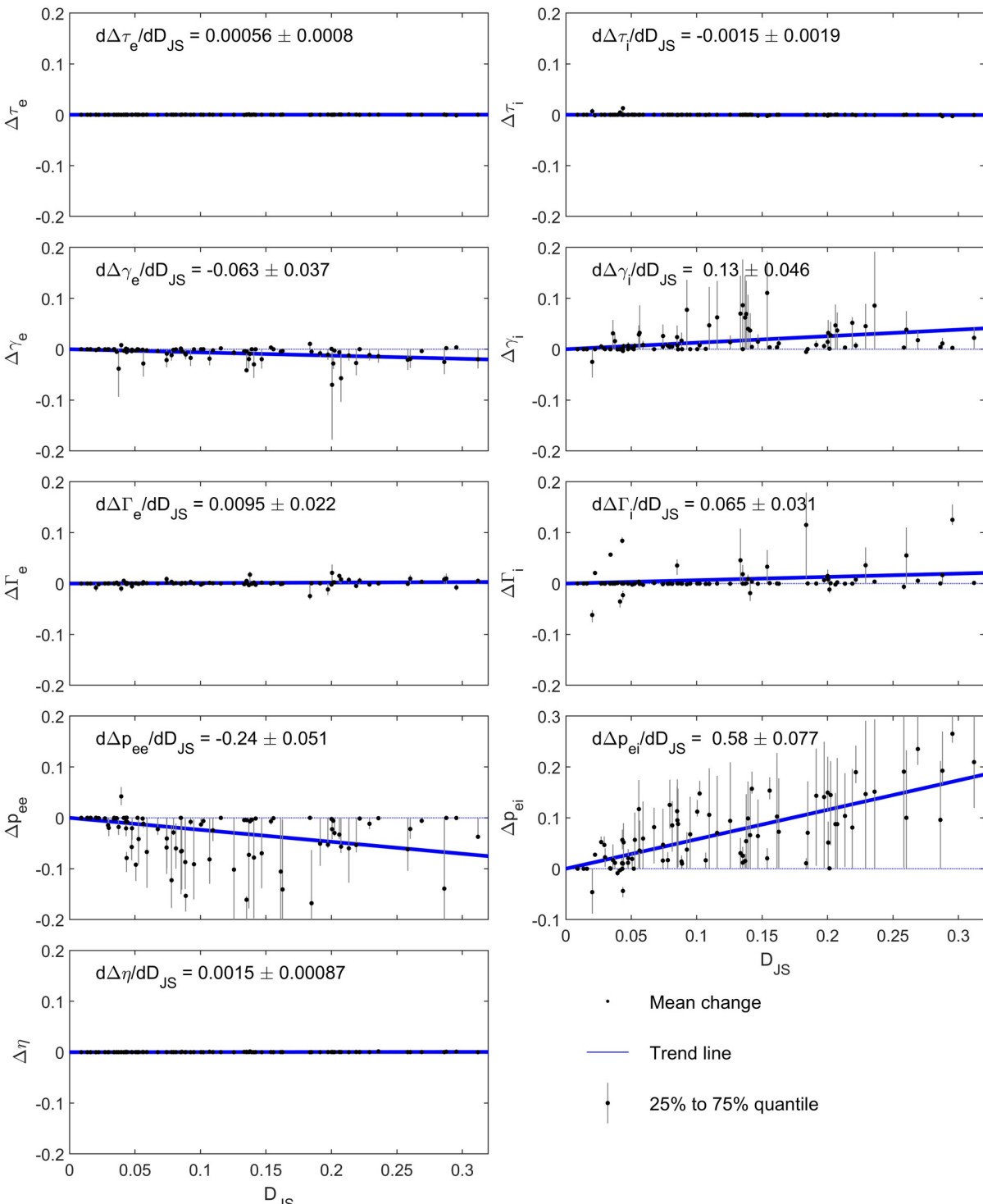

**Fig 4. EC to EO parameter responses and how they scale with the degree of alpha blocking.** The EC-to-EO parameter response (Eq 8) is calculated from the 100 best samples fits for each of the 82 subjects. The mean (black dot), calculated from Eq 9, and interquartile ranges (error bar) for each subject are plotted against the Jensen-Shannon divergence, $D_{JS}$, for that subject. In order to quantify how much each parameter response scales with the degree of alpha blocking we performed a linear regression through the sample fits; errors in the fit were estimated by randomly sampling from the distributions estimated from the sample fits. The resulting trend line is shown in blue, with its slope and error reported on each subplot. Several of the parameters ($\tau_e$, $\tau_i$, $\Gamma_e$, $\eta$) show essentially zero response to alpha blocking. Of the others, only $\Delta p_{ei}$ (lower right subplot) shows a clear trend, increasing monotonically with $D_{JS}$. $p_{ee}$ shows a non-zero parameter response but its trend with $D_{JS}$ is weak and not monotonic. This result suggests that alpha blocking by visual stimulus can largely be attributed to an increase in a tonic afferent signal $p_{ei}$ to the inhibitory cortical population, with weak or negligible contributions from the other parameters.

As a consistency check, we perform a forward calculation to test how the EEG spectrum is affected by changes in each state-distinct parameter. In Fig 5 we compare the spectra calculated from the best fit parameter set (for a particular subject), to the spectra calculated when the best-fit values for the 9 state-distinct parameters are individually perturbed. Results show that the magnitude of the alpha rhythm is most sensitive to perturbations of $p_{ei}$, with increasing $p_{ei}$

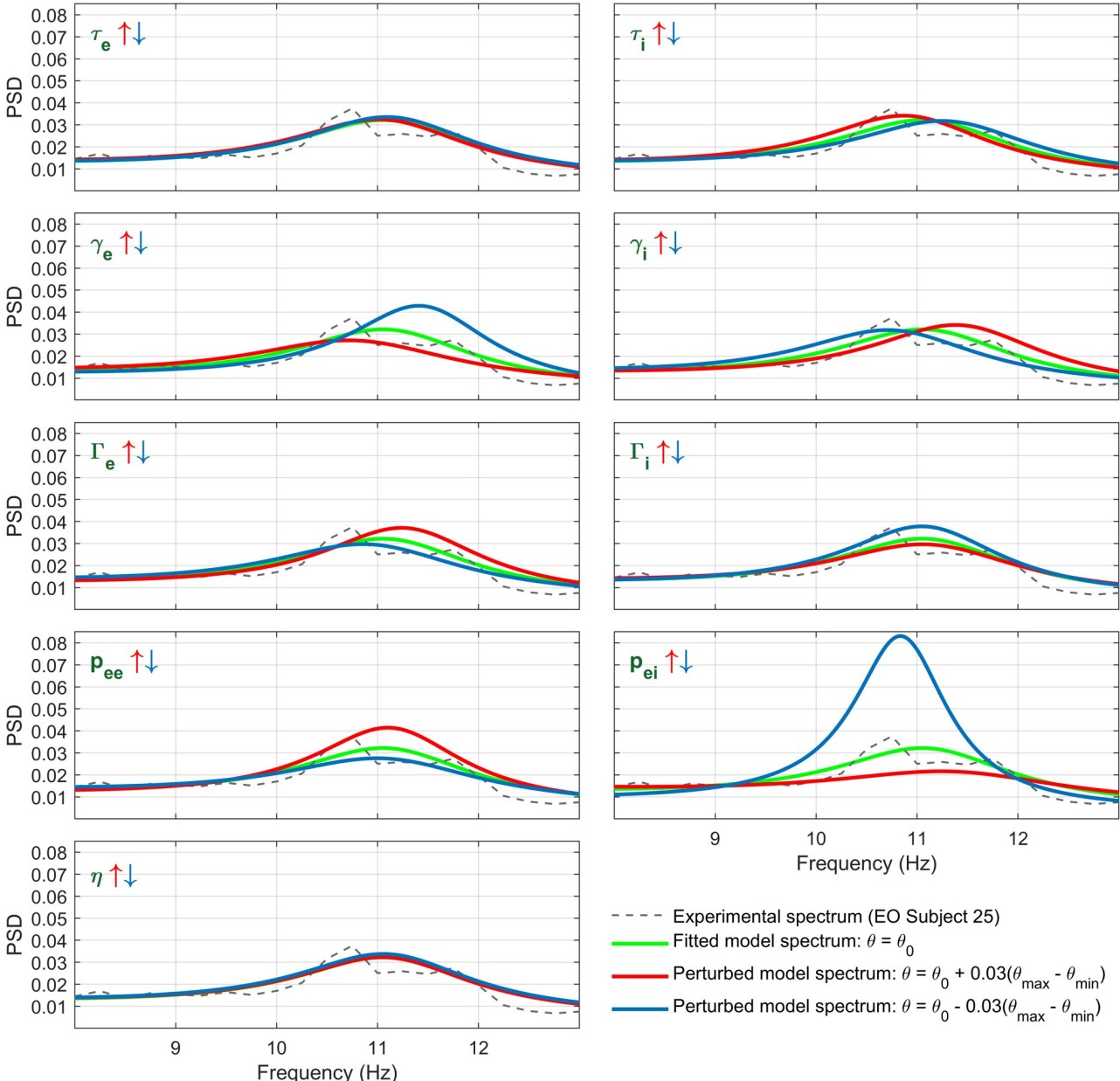

**Fig 5. Forward calculation of the sensitivity of the alpha-rhythm to individual parameters.** Shown are calculations depicting the sensitivity of the alpha-rhythm to each of the nine state-distinct parameters. The initial state (green) is that of the best fit for EO Subject 25. Each parameter is then perturbed by +3% (red) or -3% (blue) of the plausible interval, keeping other parameters constant. We observe that perturbing $p_{ei}$ changes the alpha rhythm amplitude most significantly, with a comparatively small change to the peak frequency. The same perturbations applied to $p_{ee}$ had a similar type of effect, though reversed and to a smaller extent. Alpha band power is only weakly affected by $\gamma_e$ or $\gamma_i$ though they both control the frequency. We note in general that perturbations applied to the other parameters have significantly smaller effects than perturbations to $p_{ei}$.

resulting in less alpha-band power. This is consistent with the tendency for $p_{ei}$ to increase with alpha blocking (Fig 4). Interestingly, decreases in $p_{ee}$ also cause a weaker alpha peak, although the effect is less sensitive than that for $p_{ei}$. We note that the relative effects of different parameter perturbations can vary among the different individuals, making it important to compare data across multiple individuals when performing the inverse problem.

The sensitivity of the alpha peak amplitude to changes in $p_{ei}$ helps explain why the inverse problem identified $p_{ei}$ as the dominant driver of alpha blocking: regularization is, after all, designed to identify sensitive input parameters. While this consistency is comforting, it does not rule out the role of other factors. One could, for example, contrive large changes in multiple weakly-sensitive parameters to give the same effect as a small change in a single, sensitive parameter. These are, in fact, the types of solution that a fit commonly finds without any regularization. Thus, in our effort to tame the unidentifiability problem, we are pushed towards simplicity as a guiding principle for identifying the microscopic drivers of macroscopic observations.

Importantly, we have now shown how both alpha generation and blocking can arise within a single model in a way that is justified by fits to real EEG spectra. Our previous study [33] found that the presence of spontaneous alpha oscillations was crucially dependent on the value of a single parameter—the decay rate of the inhibitory post-synaptic potential, $\gamma_i$. This confirmed the importance of *intracortical inhibition* in generating alpha activity. Our present work shows how *extra-cortical input*, particularly to inhibitory neurons, is the modulator of classical alpha blocking, making inhibition central to both the generation and modulation of alpha waves. We have thus identified the respective loci of physiological control for both the generation and attenuation of alpha oscillations.

As mentioned earlier, our model does not specify the origin of extra-cortical inputs, only that these inputs are tonic. Nevertheless, because alpha-blocking occurs throughout cortex it is reasonable to presume that these inputs are thalamo-cortical rather than long-range cortico-cortical. This is in line with previous models of thalamo-cortical dynamics [31, 32]. However, while those models invoked complex feedback between thalamus and cortex to explain alpha generation and blocking, here we claim that opening of the eyes simply alters the tonic level of thalamo-cortical afference. Thus, rather than being a driver of cortical alpha activity, the thalamus is a modulator of it.

An important feature of our results is that we find excitation of inhibitory cortical neurons to be a more sensitive modulator of the alpha rhythm than excitation of excitatory cortical neurons. This increased sensitivity to $p_{ei}$ over $p_{ee}$ arises from the state of the cortex, a cortex whose intracortical inhibition is tuned to generate spontaneous alpha oscillations. There is also anatomical evidence which indicates that thalamocortical afferents make stronger and more probable contact with inhibitory, rather than excitatory, cortical neurons [47, 48]. This means that, not only are inhibitory cortical neurons more sensitive to external inputs, they also have greater connectivity to the thalamus than do their excitatory counterparts. Both these factors indicate that thalamo-cortical excitation of inhibitory neurons is likely the dominant pathway for modulating the alpha rhythm. They also explain why opening of the eyes, which would reasonably be expected to increase thalamo-cortical input to the occipital cortex and thereby increase both $p_{ei}$ and $p_{ee}$, still causes a net attenuation of alpha activity.

In the future, the approach we have described could be used to determine the parameter response associated with anesthetic induction. Changes in EEG spectra under general anesthesia, from the loss of consciousness to the period of anaesthetic maintenance, are well characterized [49, 50]. Implementing the procedure we have outlined here, may allow us to identify a subset of the parameters driving the changes of brain state, connecting them to specific

disruptions in interneuronal communication associated with a particular anesthetic. This may provide quantitative insight into the mechanisms underlying the loss of consciousness.

## Methods

### Jensen-Shannon divergence as a measure of the degree of alpha blocking

The Jensen-Shannon divergence, $D_{JS}$, is closely related to the Kullback-Leibler divergence [51, 52]. It is symmetric, non-negative, finite, and bounded [53]. $D_{JS}$, is traditionally used to measure the difference between two probability distributions. Here we use it to measure the difference between the EC and EO spectra for each subject since these spectra have the same properties as a probability distribution: they are non-negative with a total integral of 1 (since the spectra are normalized as described in Section "EEG data"). $D_{JS}$ thus measures the difference in *shape* between EC and EO spectra, since power differences among original experimental spectra are irrelevant due to the normalization.

If the Kullback-Leibler divergence of $\boldsymbol{P}$ relative to $\boldsymbol{Q}$ is given by

$$D_{KL}(\boldsymbol{P}||\boldsymbol{Q}) = \int p(x)\ln \frac{p(x)}{q(x)}\,dx \tag{10}$$

the Jensen-Shannon divergence between the EC normalized experimental spectrum $\boldsymbol{S}^{EC}$ and the EO normalized experimental spectrum $\boldsymbol{S}^{EO}$ is given by

$$\begin{aligned} D_{JS}(\boldsymbol{S}^{EC}||\boldsymbol{S}^{EO}) \quad &= \frac{1}{2}D_{KL}(\boldsymbol{S}^{EC}||\frac{1}{2}(\boldsymbol{S}^{EC}+\boldsymbol{S}^{EO})) \\ &+ \frac{1}{2}D_{KL}(\boldsymbol{S}^{EO}||\frac{1}{2}(\boldsymbol{S}^{EC}+\boldsymbol{S}^{EO})). \end{aligned} \tag{11}$$

In this work the logarithmic base e is used in the calculation of the Jensen-Shannon divergence, in which case $0 \leq D_{JS} \leq \ln 2$.

We have chosen the Jensen-Shannon divergence based on the postulate that the greater the degree of alpha blocking, the larger the $D_{JS}$. This is qualitatively confirmed by examination of the spectra from different subjects (Fig 1 and Fig A in S1 Appendix). We are interested in how the parameter response scales with $D_{JS}$ (and thus alpha blocking), since this relates changes in the spectra to changes in the model. To check this, Fig C in S1 Appendix shows how the total parameter response (given by the Manhattan distance in parameter space, $\Sigma_m |\Delta\theta_m|$, where $m$ indexes state-distinct parameters) increases monotonically with $D_{JS}$. Distances in spectral space, captured by $D_{JS}$, thus scale smoothly with distances in parameter space, providing a link between microscopic parameters and macroscopic observables over 82 different subjects. Physiological interpretability depends, of course, on whether *individual* parameters scale with $D_{JS}$ (see Fig 4).

### State-common parameters and state-distinct parameters

There are compelling physiological reasons why certain parameters should have the same value in EC and EO states in a single individual. The EEG signal transitions reversibly between the EC and EO state in times of the order of a second. It is unlikely that parameters determined largely by the morphology or connectivity of the neurons could vary significantly on this time-scale. We thus do not expect the average number of synapses per neuron ($N_{ee}^{\beta}$, $N_{ei}^{\beta}$, $N_{ie}^{\beta}$, and $N_{ii}^{\beta}$) to vary between the two states. Similarly parameters representing intrinsic neuronal properties such as those involved in the sigmoidal response of the neural population (maximum firing rates ($S_e^{\max}$, $S_i^{\max}$) and slope ($\sigma_e$, $\sigma_i$) and the threshold ($\bar{\mu}_e$, $\bar{\mu}_i$), resting ($h_e^{rest}$, $h_i^{rest}$) and equilibrium ($h_e^{eq}$, $h_i^{eq}$) potentials) plausibly could be expected to remain constant on this time scale. We thus

require that these 14 parameters, referred to as state-common parameters, have the same value in EC and EO states for a particular individual. We emphasize that, although each of these parameters has a shared value across states, that value can vary between individuals.

The remaining 9 parameters are allowed to vary between states and are thus referred to as state-distinct parameters. Together with state-common parameters, this gives $14 + 2 \times 9 = 32$ distinct parameters down from the maximum possible $2 \times 23 = 46$ free parameters.

It is interesting to note that the state-distinct parameters fall into two sub-groups: those that characterize the input to the macro-column (i.e. tonic levels of $p_{ee}$ and $p_{ei}$ and the exponent of the input spectrum, $\eta$), and those that affect the shape, amplitude and time-scale of the post-synaptic potentials ($\gamma_e$, $\gamma_i$, $\Gamma_e$, $\Gamma_i$, $\tau_e$, and $\tau_i$). Some or all of the parameters in the state-distinct group could conceivably vary on such a time scale (though with different levels of plausibility) and so we allowed all of them to vary between states. However, after fitting and regularization, we discovered that it is primarily parameters from the first subgroup of state-distinct parameters (particularly $p_{ei}$, and to a lesser extent $p_{ee}$) that play the dominant role in distinguishing EC from EO spectra. We might regard the second sub-group as *a posteriori* shared parameters. The state-common parameters might then be referred to as *a priori* shared parameters.

## Regularization of parameter differences

A straightforward least-squares fit of EC/EO pairs resulted in parameter differences between states that showed little systematic scaling with the degree of alpha blocking (see Fig B in S1 Appendix). We hypothesized that this was caused by parameter unidentifiability (uncertainty) obscuring the subtle differences between states. To address this problem, we added a regularization term to our least-squares cost function.

Regularization is a standard method used to identify the sensitive parameters in a fit [44]. In traditional regularization, using for example the L1 norm [54], it is the value of the parameter itself that is regularized (penalized). In our case, we penalize the *differences* between (state-distinct) parameters, rather than the parameter values themselves, biasing most to zero and allowing only the most important ones to be non-zero. This reduces much of the unwanted variation caused by sloppy parameters.

The regularized cost function for the 32-parameter fit is given by

$$
\begin{aligned}
C = \quad &\frac{1}{2}\sum_n (\alpha \hat{S}_n(\boldsymbol{\theta}) S_n^{in} - S_n)^2_{EC} + \frac{1}{2}\sum_n (\alpha \hat{S}_n(\boldsymbol{\theta}) S_n^{in} - S_n)^2_{EO} \\
&+ \frac{\lambda}{N_D}\sum_{\theta_m \in \mathcal{D}} |\hat{\theta}_m^{EO} - \hat{\theta}_m^{EC}|
\end{aligned}
\tag{12}
$$

where $\boldsymbol{\theta}$ is the 32-parameter vector to be optimized, $\alpha\hat{S}(\boldsymbol{\theta})$ is the model spectrum normalized by the scaling factor $\alpha$ (the formula to compute $\alpha$ appears as Eq (12) in [33]), $S^{in}$ is the input spectrum given by $1/f^\eta$, $S$ is the experimental spectrum, $\hat{\theta}_m$ is the parameter $\theta_m$ normalized to the range of [-1,1] corresponding to the $\theta_m$'s plausible range, $\mathcal{D}$ is the set of state-distinct parameters, $N_D$ is the number of the state-distinct parameters, and $\lambda$ is the regularization parameter. The first and second terms on the right hand side correspond to least-squares fitting errors for the EC and EO spectra, respectively, while the third is the regularization term that penalizes differences between state-distinct parameters.

The amount of regularization applied (i.e. the value of $\lambda$) affects the quality of the fit. If regularization were too strong, it would force each (state-distinct) parameter to have the same value in EC and EO states, resulting in identical predicted spectra for each state and thus poor fit accuracy (assuming that the two spectra are actually different). If regularization were too

weak, the parameter values vary too wildly, as we found in Fig B in S1 Appendix. Our strategy is to maximize the amount of regularization applied while keeping the fit inside the uncertainty bounds of the data.

To determine this optimal λ, we calculate fitting errors (the first two expressions on the right hand side of Eq 12) for 19 different values of λ ranging across ten orders of magnitude. The resulting plot (see Fig D in S1 Appendix) has an "S" shape, exhibiting high fitting accuracy at low λ and poor accuracy at high λ, with a transition regime in between. Our optimal regularization parameter is taken to be the largest value of λ where the median regularised fitting error does not exceed the 84% quantile of unregularized fitting errors. This corresponds to a value λ = 0.1. A visual comparison of regularized versus unregularized fits is given in Fig 2.

## Supporting information

**S1 Appendix. Additional figures.** Fig A. Degree of alpha blocking across all subjects; Fig B. Unregularized EC to EO parameter responses and how they scale with the degree of alpha blocking; Fig C. Manhattan distances between EC and EO parameter sets as a function of the degree of alpha blocking; Fig D. Comparison of fitting error as a function of regularization parameter.
(PDF)

## Author Contributions

**Conceptualization:** Peter J. Cadusch, David T. J. Liley, Damien G. Hicks.

**Formal analysis:** Agus Hartoyo, Peter J. Cadusch, Damien G. Hicks.

**Funding acquisition:** Damien G. Hicks.

**Methodology:** Agus Hartoyo, Peter J. Cadusch, David T. J. Liley, Damien G. Hicks.

**Software:** Agus Hartoyo, Peter J. Cadusch.

**Supervision:** Peter J. Cadusch, David T. J. Liley, Damien G. Hicks.

**Visualization:** Agus Hartoyo, Peter J. Cadusch.

**Writing – original draft:** Agus Hartoyo, Peter J. Cadusch, David T. J. Liley, Damien G. Hicks.

**Writing – review & editing:** Agus Hartoyo, Peter J. Cadusch, David T. J. Liley, Damien G. Hicks.

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
