## [Decision Letter · Decision Letter 0]

25 Mar 2020

Dear Mr. Hartoyo,

Thank you very much for submitting your manuscript "Inferring a simple mechanism for alpha-blocking by fitting a neural population model to EEG spectra" for consideration at PLOS Computational Biology. As with all papers reviewed by the journal, your manuscript was reviewed by members of the editorial board and by several independent reviewers. The reviewers appreciated the attention to an important topic. Based on the reviews, we are likely to accept this manuscript for publication, providing that you modify the manuscript according to the review recommendations.

Sincerely,

Peter Neal Taylor

Associate Editor

PLOS Computational Biology

Lyle Graham

Deputy Editor

PLOS Computational Biology

[LINK]

Reviewer's Responses to Questions

**Comments to the Authors:**

Reviewer #1: This is a well-written and clear study inferring an alpha-blocking mechanism estimated by fitting EEG data with a neural population model. Overall, the results help to shed light on the still debated alpha-blocking mechanism discussion through a clear, direct and data-driven approach. The authors analyzed several EEG data corresponding to multiple individuals using a neural population model and a fitting method with the addition of a regularization term. From that, they were able to justify and find their parameter choices by fixing the ones not mutable (based on biological motivation) and by isolating the most meaningful parameter affecting the measures in the EO and EC comparison. The mechanism found corroborate with their previous study and links the alpha generation with the alpha-blocking, where the activation of the inhibitory population has an important role. Moreover, the codes and methods are available online, which is a positive point.

The paper is comprehensive and clearly written and I recommend its publication.

A few specific comments:

1) The link to access the EEG data is not working. I believe the correct is: https://archive.physionet.org/pn4/eegmmidb/ . Please, check it.

2) Although is mentioned the neuronal population model used and well-referenced, I believe that it would be more clear and easier for the reader to understand if the authors show explicitly the equations.

3) The source of the extra-cortical input (pei) and its limitations could be better discussed. Extra-cortical inputs include other sources than thalamus and, depending on the source, the cortical layers and the interneurons receiving it might be different. Moreover, it also depends if you are looking at a primary or higher-order area of the cortex. In that way, still there is no specific answer about who is driving the alpha-blocking. Enriching this discussion will clarify the limitations and the possible ways to test it through experiments and more detailed models.

Reviewer #2: The authors construct a firing rate/population model of multiple neural populations (E, I) and fit

the spectra generated by the model to EEG data from 5 subjects with eyes open and closed

conditions. They use a particle swarm optimization to fit the different conditions and find

that in their model the excitatory inputs to the interneuron population is the major determinant of

alpha reduction in the EO condition. Overall, the writing is clear, and the results will

be of interest to the neuroscience community. In line with current practices, the authors

have shared their source code.

While compelling, the authors should more clearly explain a neuroanatomical rationale for why

only inputs to the interneurons regulate alpha. Is there a thalamic source for this? Could

it be provided via thalamic matrix inputs (Biological Psychiatry 87(8):770)? Authors should cite

relevant literature.

In addition, the authors should discuss whether the type of model they developed has

enough biological detail to offer novel insights into mechanisms of brain rhythm generation

and their modulation. Many detailed circuit models and modeling platforms are now

available that have competing explanations for the origin of alpha (e.g. see eLife. 2020; 9:

e51214.). Authors should compare their model against some of these other models/tools.

As far as organization of the manuscript, the authors should move the figures in Discussion

into the Results, along with the description of those figures.

Detailed comments:

field models - external input to inhibitory neurons in cortex responsible for attenuating alpha

they fit EEG data with eyes open (alpha higher) and closed (alpha lower) using population model

and found that one parameter - external input to inhibitory neurons in cortex was responsible

for modulating alpha power. that's not so surprising - but what is the explanation? does it fit

the neuroanatomical data? mechanistic models?

why would opening eyes increase drive to cortical ihibitory neurons? which pathway is

responsible?

there are many models that can account for the data ...

105-108:

"Local equations are linearized around a fixed point and the power

spectral density (PSD) is derived assuming a stochastic driving signal of the excitatory

population that represents thalamo-cortical and long range cortico-cortical inputs,

assumed to be Gaussian white noise. The modelled PSD can then be written as a"

Why is thalamocortical drive assumed to be white noise? Is that realistic

given knowledge of thalamocortical dynamics? I would think that some peaks

in frequency, e.g. in alpha range would be more realistic.

OK, then later they mention that the inputs are not white noise, so that's

a fittable parameter that influences the noise type provided (white, pink, brown, etc.).

plos comp bio thalamic model - more realistic and offers more plausible insights

into mechanisms of rhythm generation

DJS - nice measure for quantifying differences in power spectra

Fig.3 may have too much detail for the typical reader. Is there a way to summarize the

fitted distributions for each patient rather than displaying 23 x 5 distributions??

Line 188-190: if most parameter responses are 0 or insignificant trend with degree of alpha

blocking, why not instead show the parameter response that are significant or not 0??

The beginning of the Discussion and Figures 4 and 5 should be moved into the Results.

Although the discussion around lines 221 address some of this,

can the authors comment on the mechanism as to why the parameter p_ei (excitatory

input to inhibitory neurons) is the major determinant of changes in alpha

between the EO and EC conditions and why p_ee is not important? I would have

thought both parameters should influence the magnitude of oscillations. In addition,

which neuroanatomical pathway would set the p_ei value and how would that pathway

influence only the interneurons? Is the model-predicted parameter influencing

alpha consistent with experimental data?

**Have all data underlying the figures and results presented in the manuscript been provided?**

Reviewer #1: Yes

Reviewer #2: Yes

PLOS authors have the option to publish the peer review history of their article (what does this mean?). If published, this will include your full peer review and any attached files.

Reviewer #1: No

Reviewer #2: No
---

## [Editor Report · Decision Letter 1]

7 Apr 2020

Dear Hartoyo,

We are pleased to inform you that your manuscript 'Inferring a simple mechanism for alpha-blocking by fitting a neural population model to EEG spectra' has been provisionally accepted for publication in PLOS Computational Biology.

Best regards,

Peter Neal Taylor

Associate Editor

PLOS Computational Biology

Lyle Graham

Deputy Editor

PLOS Computational Biology

---

## [Editor Report · Acceptance letter]

22 Apr 2020

PCOMPBIOL-D-20-00063R1 

Inferring a simple mechanism for alpha-blocking by fitting a neural population model to EEG spectra

Dear Dr Hartoyo,

I am pleased to inform you that your manuscript has been formally accepted for publication in PLOS Computational Biology. Your manuscript is now with our production department and you will be notified of the publication date in due course.

With kind regards,

Sarah Hammond
